# Prostate Dimensions and Their Impact on LUTS and Erectile Function: Is Length the Missing Link?

**DOI:** 10.3390/jcm13237123

**Published:** 2024-11-25

**Authors:** Daniel Porav-Hodade, Mihai Dorin Vartolomei, Toader Septimiu Voidazan, Raul Gherasim, Iulia Andras, Ciprian Todea-Moga, Bogdan Ovidiu Feciche, Silvestru-Alexandru Big, Mártha Orsolya Katalin Ilona, Ioan Coman, Nicolae Crisan

**Affiliations:** 1Department of Urology, George Emil Palade University of Medicine, Pharmacy, Science, and Technology of Târgu Mureș, 540139 Târgu Mureș, Romania; daniel.porav-hodade@umfst.ro (D.P.-H.); orsolya.martha@umfst.ro (M.O.K.I.); raul-dumitru.gherasim@umfst.ro (R.G.); 2Department of Urology, Clinical County Hospital Mureș, 540136 Târgu Mureș, Romania; 3Antares Clinic, 610005 Piatra Neamt, Romania; 4Department of Cell and Molecular Biology, George Emil Palade University of Medicine, Pharmacy, Science, and Technology of Târgu Mureș, 540139 Târgu Mureș, Romania; mihai.vartolomei@umfst.ro; 5Department of Epidemiology, George Emil Palade University of Medicine, Pharmacy, Science, and Technology of Târgu Mureș, 540139 Târgu Mureș, Romania; septimiu.voidazan@umfst.ro; 6Department of Urology, Iului Hatieganu University of Medicine and Pharmacy, 400129 Cluj-Napoca, Romania; dr.iuliaandras@gmail.com (I.A.); jcoman@yahoo.com (I.C.); nicolae.crisan@umfcluj.ro (N.C.); 7Department of Urology, Faculty of Medicine and Pharmacy, University of Oradea, 410087 Oradea, Romania; 8Department of Urology, Emergency County Hospital Oradea, 410169 Oradea, Romania; alex_big_jsd@yahoo.ro

**Keywords:** LUTS, erectile function, IPSS, IIEF, prostate volume, length of the prostate, hypertension, diabetes mellitus, cardiovascular diseases, smoking, alcohol consumption, PSA

## Abstract

**Background/Objectives**: The objective of this study is to explore potential correlations between prostate volume, LUTS, and IIEF, with a particular emphasis on the relationship between prostate dimensions—width, height, and length—and both LUTS and IIEF and to assess patients based on risk factors such as hypertension, diabetes, cardiovascular disease, smoking, alcohol consumption, and PSA levels. **Methods**: A retrospective multicenter study was conducted between January 2007 and December 2023, focusing on male patients over the age of 40. The study evaluated hypertension, diabetes, cardiovascular diseases, smoking, alcohol consumption, and lower urinary tract symptoms (LUTS) through the completion of the IPSS and QoL questionnaires, sexual function using the IIEF-15, and PSA levels. Abdominal ultrasound was performed to determine prostate volume and its dimensions (width, height, and length). **Results**: A total of 943 patients were included in the study, with a mean age of 61.89 ± 8.51 years. From the 40–49 age group to the 80–90 age group, IPSS increased from 10.29 to 14.26 points, PSA from 1.1 ng/mL to 3.05 ng/mL, and prostate volume from 23.79 mL to 41.16 mL. Meanwhile, over the same age intervals, IIEF showed a decline from 52.57 to 24.76 points. The IPSS demonstrated a statistically significant positive correlation (*p* < 0.05) with prostate volume and patient age, while showing an inverse correlation with IIEF. The only statistically significant correlation between IPSS and prostate dimensions was with the length diameter of the prostate (*p* = 0.011). The severity of sexual symptoms was inversely correlated with both prostate volume and age. Additionally, IIEF was negatively correlated with the width and length diameters of the prostate. Hypertension (*p* = 0.57), diabetes (*p* = 0.57), smoking (*p* = 0.76), and alcohol consumption (*p* = 0.27) did not have a statistically significant impact on IPSS, and IIEF except for cardiovascular diseases, which showed a significant correlation with IPSS in patients experiencing moderate to severe symptoms (*p* = 0.0001). The statistically significant correlation between cardiovascular diseases and IIEF was observed only in patients with severe symptoms (*p* = 0.0001). **Conclusions**: There is a correlation between prostate volume, IPSS, and IIEF. Only length of the prostate shows a statistically significant correlation with both IPSS and IIEF. PSA levels increase progressively with each decade of age. Hypertension, diabetes, smoking, and alcohol consumption do not have a statistically significant impact on LUTS and erectile function. Cardiovascular diseases show a correlation with patients experiencing moderate to severe LUTS, as well as with those who have severe symptoms according to the IIEF evaluation.

## 1. Introduction

Lower urinary tract symptoms (LUTS) are a frequent issue among adult men, particularly those over the age of 50, and are closely linked to aging. They significantly affect quality of life (QoL) and pose a considerable economic burden [1]. The International Prostate Symptom Score (IPSS) is widely used to assess the severity of LUTS. These symptoms are often associated with benign prostatic hyperplasia (BPH). Additionally, lower urinary tract symptoms are linked to several modifiable risk factors, highlighting potential opportunities for prevention. The relationship between systemic conditions such as hypertension (HTN), diabetes mellitus (DM), and cardiovascular diseases (CVD) with lower urinary tract symptoms and erectile dysfunction (ED) has been well documented [2,3].

Smoking and alcohol consumption are the most implicated lifestyle factors in the majority of pathologies, and LUTS is no exception. However, the results of studies regarding their effects on LUTS are inconsistent, ranging from the protective action of alcohol to its negative impact on BPH/LUTS [4].

Additionally, erectile dysfunction (ED) is another frequent condition in aging men. Erectile dysfunction is characterized by the consistent inability to achieve and sustain an erection adequate for satisfactory sexual activity, significantly affecting the quality of life for both the patient and their partner [5,6]. Epidemiological studies have demonstrated a high global prevalence and incidence of erectile dysfunction [5]. It can be assessed using validated psychometric questionnaires, such as the International Index of Erectile Function (IIEF), specifically the IIEF-15 questionnaire [7].

Numerous risk factors are implicated in the etiology of ED. In addition to age, conditions such as diabetes mellitus, hypertension, cardiovascular disease, obesity, and metabolic syndrome (MetS) are associated with ED [8,9]. Smoking and drug use are lifestyle factors that impact ED, although there is no consensus regarding the effect of alcohol consumption [10].

Studies show a notable correlation between LUTS and ED, particularly in men over 50 years of age [11]. Prostate volume is one of the factors that correlates with both LUTS and erectile dysfunction [12,13]. However, there are no studies that show whether certain prostate dimensions have a stronger or lesser impact on urinary symptoms or erectile function.

The objective of this study is to explore potential correlations between prostate volume, LUTS, and IIEF, with a particular emphasis on the relationship between prostate dimensions—width, height, and length—and both LUTS and IIEF. Furthermore, the study aims to assess patients based on risk factors such as hypertension, diabetes, cardiovascular disease, smoking, alcohol consumption, PSA levels, and identify any statistically significant correlations between these factors, LUTS, and IIEF.

## 2. Materials and Methods

### 2.1. Participant Selection

This retrospective multicenter study was conducted between January 2007 and December 2023, including a total of 943 patients.

#### 2.1.1. Inclusion Criteria

The inclusion criteria were male patients over the age of 40, consent to a comprehensive evaluation related to hypertension, diabetes, cardiovascular diseases, smoking, alcohol consumption, lower urinary tract symptoms (LUTS) (completion of the International Prostate Symptom Score, and Quality of Life questionnaires), symptoms related to sexual dysfunction (completion of the International Index of Erectile Function–15). Consent for performing an ultrasound to determine prostate size, as well as conducting a PSA test, was also a mandatory requirement.

All patients included in the group were treatment-naïve for LUTS or erectile dysfunction.

For the evaluation of hypertension, diabetes, cardiovascular diseases, smoking, and occasional alcohol consumption, patients answered a questionnaire with yes/no options, without assessing the severity of the conditions or the medication used, the number of cigarettes smoked per day, or the volume and type of alcohol consumed.

The International Prostate Symptom Score (IPSS) is an eight-question survey that includes seven questions about symptoms and one question addressing quality of life (QoL). The scoring is classified as follows: asymptomatic (0 points), mild symptoms (1–7 points), moderate symptoms (8–19 points), and severe symptoms (20–35 points) [14]. In the final statistical analysis, asymptomatic patients were included in the group of patients with mild symptoms (42 patients).

The International Index of Erectile Function is a validated psychometric questionnaire used to assess sexual function. It comprises 15 questions that evaluate erectile function, orgasmic function, sexual desire, intercourse satisfaction, and overall sexual satisfaction. In the final statistical evaluation, all 15 questions were used, each with 5 possible answers (1–5 points per question), not just those related to erectile function (Q1–5, 15). Patients were classified into several groups based on the severity of sexual dysfunction: no dysfunction (61–75 points), mildly symptomatic (41–60 points), moderately symptomatic (21–40 points), and severely symptomatic (≤20 points) [7].

Abdominal ultrasound for measuring prostate dimensions was performed using a 3.5 MHz abdominal transducer. The prostate volume was calculated by measuring its three dimensions: in the transverse plane (cross-sectional view), the width (D1) was measured from side to side, and in the same plane, the height from (D2) the anterior (front) to the posterior (back) aspect was recorded. Finally, in the sagittal plane, the length (D3) of the prostate from the base to the apex (top to bottom) was measured. The prostate volume was calculated using the following formula: prostate volume (mL) ≈ 0.52 × length × width × height [15].

#### 2.1.2. Exclusion Criteria

The exclusion criterion was the patient’s lack of consent for the evaluation described above. Additionally, the study excluded the COVID-19 period from January 2020 to December 2021, as no data from that time could be included in the study.

All patients who did not have complete data for all variables included in this study were excluded from the final statistical analysis.

The study was approved by the Ethics Committee of the Clinical County Hospital Mures, 540136 Târgu Mureș, Romania (13735/09.09.2024), Romania, and conducted in accordance with the Declaration of Helsinki.

### 2.2. Statistical Analysis

Data were considered as nominal or quantitative variables. Nominal variables were characterized using frequencies. Quantitative variables were tested for normality of distribution using Kolmogorov–Smirnov test and were characterized by median and minimum–maximum or by mean and standard deviation (SD), when appropriate. A chi-square test was used in order to compare the frequencies of nominal variables. Quantitative variables were compared using *t* test, Mann–Whitney test, ANOVA test (Bonferroni correction), or Kruskal–Wallis test (Dunns correction), when appropriate. The correlation between quantitative variables was assessed using Pearson correlation or Spearman’s rho, when appropriate. The level of statistical significance was set at *p* < 0.05. Statistical analysis was performed using SPSS for Windows version 23.0 (SPSS, Inc., Chicago, IL, USA). Multivariate analysis was carried out using linear regressions. We used as dependent variable the IPSS or IIEF, and independent variable: age, D1, D2, D3, PSA, volume.

## 3. Results

Out of the total of 943 patients, 923 met all inclusion and exclusion criteria. The mean age was 61.89 ± 8.50 (Table 1).

PSA values ranged from 0.01 ng/mL to 41 ng/mL, with a median value of 1.2 ng/mL. The average PSA values by age decade were as follows: 1.11 ± 0.88 ng/mL for ages 40–49, 1.19 ± 0.90 ng/mL for ages 50–59, 1.93 ± 2.16 ng/mL for ages 60–69, 2.83 ± 4.62 ng/mL for ages 70–79, and 3.05 ± 2.65 ng/mL for patients over 80.

Hypertension was the most prevalent comorbidity, affecting 46.34% of the 943 patients included in the study. Alcohol consumption was the most commonly reported toxic habit, with 26% of patients mentioning it. (Table 2).

Evaluating the LUTS symptoms of the patients included in the study using the IPSS test, 38.8% were asymptomatic (IPSS ≤ 7), 42.2% presented with moderate symptoms (IPSS = 8–19), and 18.82% had severe symptoms (IPSS ≥ 20).

Regarding the quality of life related to urinary symptoms (QoL), 9.2% of patients reported being delighted, 20.6% pleased, 7.2% mostly satisfied, 15.1% mixed, 16.5% mostly dissatisfied, 7.7% unhappy, and 2.4% described their condition as terrible. A significant percentage of patients (21.3%) did not respond to this question.

An analysis of the group based on sexual function (IIEF-15) showed that only 14.5% of patients exhibited normal sexual activity, while nearly half (45.8%) experienced moderate sexual dysfunction (Table 3).

A descriptive analysis of the patient group by age decades reveals a direct proportional relationship between age and IPSS, PSA, and prostate volume, prostate diameters, while the relationship with IIEF was inversely proportional.

From the 40–49 age group to the 80–90 age group, IPSS increased from 10 to 14 points, PSA from 1.1 ng/mL to 3.05 ng/mL, and prostate volume from 23.79 mL to 41.16 mL. Meanwhile, over the same age intervals, IIEF showed a decline from 53 to 25 points (Table 4).

The distribution of patients with moderate or severe LUTS symptoms by age group was as follows: 52.8% for those aged 50–59, 60.8% for the 60–69 age group, 75.4% for patients aged 70–79, and 70.5% for those over 80 years old.

The same decade-based trend was observed for IIEF. We applied nonparametric Spearman correlations (since IPSS does not follow a Gaussian distribution) to assess the presence of statistically significant correlations between IPSS and the studied parameters.

Prostate symptoms show a statistically significant direct correlation with the prostate volume, and patient age, and an inverse correlation with IIEF. Although IPSS generally shows a statistical correlation with prostate volume, regarding prostate diameters, the only statistically significant correlation is with the length diameter (D3) of the prostate, from the base to the apex.

With increasing severity of LUTS symptoms, the IIEF score decreases, indicating a rise in the severity of sexual dysfunction (*p* < 0.05) (Table 5).

We used parametric Pearson correlations (as IIEF follows a Gaussian distribution) to assess the relationships between IIEF, prostate dimensions, and patient age.

The severity of sexual symptoms shows an inverse correlation with prostate volume and age. Furthermore, IIEF is negatively correlated with diameters D1 and D3 (*p* < 0.05) (Table 6).

The statistical analysis using chi-square tests to assess the impact of hypertension (*p* value = 0.57), diabetes (*p* value = 0.57), cardiovascular diseases (*p* value = 0.0001), smoking (*p* value = 0.76), and alcohol consumption (*p* value = 0.27) on prostate symptoms (IPSS) reveals that, except for cardiovascular diseases, none of the other factors had a statistically significant impact on IPSS. Regarding cardiovascular diseases, the statistically significant correlation between patients with cardiovascular diseases and IPSS was observed in patients with moderate and severe symptoms (Table 7).

The same type of statistical analysis was applied for IIEF. We performed a statistical analysis using chi-square tests to evaluate the impact of hypertension (*p*-value = 0.71), diabetes (*p*-value = 0.24), cardiovascular diseases (*p*-value = 0.0001), smoking (*p*-value = 0.36), and alcohol consumption (*p*-value = 0.28) on sexual function (IIEF). Except for cardiovascular diseases, none of the factors mentioned above had a statistically significant impact on IIEF. The statistically significant correlation between cardiovascular diseases and IIEF was found in patients with severe symptoms (IIEF < 20) (Table 8).

In the multivariate regression model, the IPSS is taken as a dependent variable. IPSS is positively influenced by age, D3, and volume. The same IIEF is taken as a dependent variable in the multivariate regression model (Table 9). IIEF is negatively influenced by age, D3, and volume (Table 10).

## 4. Discussion

This retrospective study primarily aimed to investigate the correlation between prostate volume, prostate dimensions, and BPH/LUTS symptoms, as well as IIEF.

Our study demonstrates that with increasing age, there is both a significant rise in the prevalence of LUTS and a decline in erectile function, evidenced by a reduction in IIEF scores.

For each decade of age, prostate-related symptoms progressively worsened, starting with a median IPSS of 10.29 ± 7.1 in the 40–49 age group and increasing to 14.06 ± 8.85 in the 80–89 age group. Conversely, IIEF scores decreased from a median of 52.57 ± 13.57 in the 40–49 age group to 24.76 ± 16.32 in the 80–89 age group. These findings align with those in the literature. Haidinger et al. [16] conducted a large-scale cross-sectional study that highlights the critical role of age in the development of LUTS. Other studies have also demonstrated this association [17,18].

Age similarly has a negative impact on erectile function, with the IIEF score gradually declining to an average of 24.76 ± 16.32 in the 80–90 age decade.

In our study, there was also a statistically significant correlation between IPSS and IIEF. Indeed, the relationship between age, IPSS, and IIEF is well-known and has been demonstrated in various studies [19,20,21].

For the entire study group, we observed an increase in PSA levels correlated with age decades. PSA levels peaked in the 80–89 age group, with a median of 3.05 ± 2.65 ng/mL, while the lowest levels were recorded in the 40–49 age group, with a median of 1.11 ± 0.88 ng/mL.

Prostate volume is strongly related to serum PSA in men with BPH, with serum prostate-specific antigen serving as an excellent predictor of prostate volume [22]. Furthermore, in a multicenter study evaluating the relationship between serum prostate-specific antigen and prostate volume, Chung et al. [23] obtained similar findings regarding PSA levels by age decade (PSA levels of >1.3 ng/mL, >1.7 ng/mL, and >2.0 ng/mL for men with BPH in their sixth, seventh, and eighth decades, respectively).

The multifactorial nature of LUTS and erectile dysfunction pathogenesis makes it challenging to determine the individual role of each contributing factor. Endothelial dysfunction in the pelvic vascular system likely plays a central role in the pathogenesis of both conditions [24]. There is no evidence that LUTS or BPH directly causes cardiovascular disease. However, CVD, through mechanisms such as ischemia or chronic inflammation, may lead to LUTS via BPH [25]. In a study by Wehrberger et al. [26] involving patients aged 30 to 92, it was shown that men with severe LUTS (IPSS ≥ 20) had an increased risk for CVD, whereas those with mild to moderate LUTS did not show a statistically significant correlation with CVD.

The link between erectile dysfunction (ED) and cardiovascular disease (CVD) is significantly better established compared to its association with LUTS/BPH. Furthermore, ED is increasingly recognized as an early marker for CVD [27]. This interconnection is largely established through endothelial dysfunction, atherosclerotic plaque formation, hormonal factors (such as low testosterone), as well as chronic inflammation and oxidative stress [28,29,30]. Both conditions share common risk factors, including hypertension, diabetes, obesity, and smoking [27].

Although it is generally accepted that men with hypertension tend to have higher IPSS scores and larger prostate volumes than those without hypertension [31], and that hypertension raises the risk of erectile dysfunction (ED)—with ED often serving as an early indicator of hypertension [32]—in our study, this correlation, while present, did not reach statistical significance.

The situation was similar for diabetes mellitus. In a meta-analysis, Xin et al. [33] concluded that LUTS in BPH patients is more pronounced among those with diabetes mellitus compared to controls. Additionally, the long-term complications of diabetes, including macroangiopathy, microangiopathy, and neuropathy, can impact erectile function, with diabetic patients facing a 3.5-fold higher risk of developing ED compared to non-diabetic individuals [34]. However, in our study, this correlation did not reach statistical significance.

In addition to age, conditions such as diabetes mellitus, hypertension, cardiovascular disease, and obesity, and lifestyle factors like smoking and alcohol consumption, have been extensively studied for their effects on various health conditions, including lower urinary tract symptoms (LUTS) and erectile dysfunction (ED). However, there is no consensus on the specific impact, particularly regarding alcohol consumption.

Study results regarding the effects of alcohol and smoking on LUTS remain inconsistent, ranging from alcohol’s protective effects to its negative impact on BPH/LUTS. Noh et al. [35], investigating the association between LUTS and cigarette smoking or alcohol consumption, concluded that alcohol intake positively affected daytime LUTS but had a negative impact on nighttime symptoms. In contrast, cigarette smoking showed the opposite trend, with a negative effect on daytime LUTS and a positive effect on nocturia. Meanwhile, in the NHANES III study, Rohrmann et al. [36] found that moderate alcohol consumption provided a protective effect against LUTS, while cigarette smoking was not associated with LUTS.

Alcohol’s impact on erectile function varies with the amount and duration of consumption. While moderate amounts can have a positive effect through disinhibition and relaxation, chronic consumption negatively impacts erectile function by potentially causing vascular damage [37].

Evidence from observational studies suggests that smoking is significantly associated with erectile dysfunction, this association being dependent on both the duration and quantity of cigarettes consumed. [10,38]. Recent studies have shown that e-cigarette use may contribute to ED, with endothelial damage likely serving as a potential mechanism for this effect [39]. The impact of e-cigarettes on lower urinary tract symptoms (LUTS), however, is not yet fully understood [40].

In our group, lifestyle factors, such as smoking and alcohol consumption, did not show a statistically significant impact on either IIEF or IPSS.

As previously mentioned, the primary objective of this study was to evaluate potential correlations between prostate volume, LUTS, and IIEF, with a particular emphasis on determining whether certain prostate dimensions have a more significant impact on LUTS and/or IIEF.

In our study, prostate volume was found to have a statistically significant correlation with LUTS. However, the complex relationship between LUTS and prostate volume remains a topic of debate in urology. While earlier studies suggested that prostate size does not correlate with bladder outlet obstruction [41,42], more recent studies increasingly support this association [43]. Rehman et al. [44] concluded that there is a significant positive correlation between the International Prostate Symptom Score (IPSS) and prostate volume in patients with BPH. Similarly, a recent study by Kim et al. [45] reached the same conclusion, finding a relationship between prostate volume and LUTS, albeit a low correlation. Despite the numerous studies supporting this association, consensus on this correlation remains elusive.

There are currently no studies that individually assess the impact each prostate dimension may have on LUTS. In our study cohort, only the length (D3) of the prostate showed a statistically significant correlation with IPSS. Although correlations were observed for the other two dimensions, they were not statistically significant.

Existing studies primarily focus on prostate length (base–apex diameter) and the significance of intravesical prostatic protrusion (IPP) into the bladder [46,47]. There is consensus among these studies on the significant direct correlation between prostate length and IPSS, with intravesical prostatic protrusion regarded as an independent risk factor for the severity of bladder outlet obstruction [48].

The degree of IPP can be measured from the intravesical edge of the prostate to the base of the bladder in the mid-sagittal abdominal ultrasound [49].

However, IPP measurement was not initially considered as part of this study’s design, though this assessment was later performed for some patients. As a result, we did not evaluate the impact of IPP on IPSS statistically and, consequently, cannot determine whether there is a positive relationship even when IPP is not severe. We recognize that this omission limits our findings, and we consider it one of the study’s limitations.

Regarding the correlation between prostate volume, its dimensions, and IIEF, our study found a statistically significant relationship between prostate volume and IIEF, as well as between the length diameter (D3) of the prostate and IIEF. Additionally, for these patients, the width diameter (D1) of the prostate was also significantly correlated with IIEF. Kardasevic and Milicevic [50], using IIEF-5 to assess erectile dysfunction, concluded that there is an inverse relationship between erectile dysfunction and prostate volume, where an increase in prostate volume leads to a decline in IIEF score [13].

Qalawena et al. [51] using penile Doppler ultrasound to assess the vascular characteristics of erection, including peak systolic velocity, end diastolic velocity, and resistive index (RI), concluded that there is a significant correlation between prostate volume and IIEF. Specifically, an increased transitional zone volume of the prostate was associated with decreased IIEF.

Currently, there are no studies that specifically correlate the three dimensions of the prostate and their individual impact on erectile function.

Theories regarding the physiological and pathophysiological causes linking prostate length to prostate symptomatology (IPSS) and erectile dysfunction (IIEF) include several potential mechanisms worth discussing. As the prostate enlarges in its cranio-caudal (length) dimension, it may exert increased pressure on the prostatic urethra, leading to bladder outlet obstruction. This urethral compression increase urinary resistance, contributing to voiding symptoms and elevated IPSS scores.

Similarly, an elongated prostate may compress adjacent neurovascular bundles, potentially reducing blood flow to the penile vasculature or disrupting nerve pathways, both of which could impair erectile function.

Additionally, lengthening of the prostate may influence the autonomic nervous system, which governs involuntary functions like smooth muscle contraction in the prostate and bladder. This enlargement could overstimulate adrenergic nerves, leading to increased smooth muscle tone in the bladder neck and prostate. Such sympathetic overactivity may further impair erectile function by inhibiting the parasympathetic pathways necessary for an erection.

Of course, these are theories we have considered, and continued research is essential to clarify the possible shared pathophysiological mechanisms between LUTS and erectile dysfunction.

The strengths of this study include a large patient sample, an innovative approach to correlating specific prostate dimensions with both IIEF and IPSS scores, and the examination of IPSS and IIEF changes across age groups. Additionally, it evaluates other key parameters, such as cardiovascular disease and diabetes, as well as lifestyle risk factors like smoking and alcohol consumption.

The main limitations involve the use of abdominal ultrasound for prostate measurement, which is less precise than transrectal ultrasound; however, abdominal ultrasound was chosen for consistency throughout the study. Another limitation is the lack of detailed erectile function analysis based on the first five questions of the IIEF, as only the IIEF-15 total score was included in the dataset, limiting a more specific correlation.

## 5. Conclusions

There is a correlation between prostate volume, IPSS, and IIEF. Only the cranio-caudal diameter (length) of the prostate shows a statistically significant correlation with both IPSS and IIEF. PSA levels increase progressively with each decade of age. Hypertension, diabetes, smoking, and alcohol consumption do not have a statistically significant impact on LUTS and erectile function. Cardiovascular diseases show a correlation with patients experiencing moderate to severe LUTS, as well as with those who have severe symptoms according to the IIEF evaluation. Continued research is essential to clarify the pathophysiological mechanism by which prostate length may impact LUTS and sexual function.

## Figures and Tables

**Table 1 jcm-13-07123-t001:** Cohort demographics (age, IPSS, IIEF, prostate diameters, and volume).

	Age	IPSS	IIEF	D1	D2	D3	Volume
Mean ± SD	61.89 ± 8.50	11 ± 8	43 ± 16	38.75 ± 7.18	39.22 ± 7.51	37.27 ± 6.77	31.10 ± 15.73
N	923	923	923	923	923	923	923

**Table 2 jcm-13-07123-t002:** Prevalence of comorbidities and risk factors (smoking, alcohol consumption) among the patients included in the study.

	Hypertension	Diabetes Mellitus	Cardiovascular Disease	Smoking	Alcohol Consumption
Nr (%)	435 (47.12)	89 (9.65)	201 (21.77)	148 (16.03)	239 (25.89)
Nr total	923	923	923	923	923

**Table 3 jcm-13-07123-t003:** Prevalence of comorbidities and risk factors (smoking, alcohol consumption) among the patients included in the study.

IIEF-15	Frequency	Percent
IIEF = 61–75	131	14.19
IIEF = 41–60	425	46.05
IIEF = 21–40	212	22.97
IIEF ≤ 20	155	16.79

**Table 4 jcm-13-07123-t004:** Changes in IPSS, IIEF, prostate volume, and prostate diameters reported by decades.

Age Group (years)		IPSS	IIEF	Volume	D1 Width	D2 Height	D3 Length
40–49	Mean ± SD	10 ± 7	53 ± 14	23.79 ± 10.70	35.12 ± 6.66	36.75 ± 6.02	33.16 ± 8.27
	N	48	48	48	48	48	48
50–59	Mean ± SD	10 ± 8	49 ± 15	26.51 ± 9.65	36.65 ± 7.19	37.25 ± 7.38	34.62 ± 7.75
	N	335	335	335	335	335	335
60–69	Mean ± SD	12 ± 8	42 ± 14	32.73 ± 15.70	39.23 ± 7.23	39.8 ± 7.87	37.73 ± 7.61
	N	360	360	360	360	360	360
70–79	Mean ± SD	13 ± 8	29 ± 14	37.92 ± 20.70	41.21 ± 8.14	41.30 ± 7.97	39.64 ± 7.77
	N	163	163	163	163	163	163
80–90	Mean ± SD	14 ± 9	25 ± 16	41.16 ± 29.66	42.9 ± 8.80	42.23 ± 12.54	39.65 ± 8.59
	N	17	17	17	17	17	17
Total	N	923	923	923	923	923	923

**Table 5 jcm-13-07123-t005:** Spearman correlations between IPSS, sexual function, age, and prostate dimensions.

			IPSS	IIEF-15	D1	D2	D3	Volume	Age
Spearman’s rho	IPSS	Correlation coefficient rho	-	−0.22 **	0.058	0.064	0.084 *	0.086 **	0.151 **
		Sig. (two-tailed-*p* value)	-	0.000	0.082	0.052	0.011	0.010	0.000
	IIEF-15	Correlation coefficient rho	−0.219 **	-	−0.132 **	−0.053	−0.111 **	−0.114 **	−0.464 **
		Sig. (two-tailed-*p* value)	0.000	-	0.000	0.0106	0.001	0.001	0.000
	Age	Correlation coefficient rho	0.151 **	−0.46 **	0.241 **	0.211 **	0.291 **	0.302 **	-
		Sig. (two-tailed-*p* value)	0.000	0.000	0.000	0.000	0.000	0.000	-

** Correlation is significant at the 0.01 level (two-tailed). * Correlation is significant at the 0.05 level (two-tailed). (+) Plus sign in the correlation coefficient indicates a positive correlation, meaning both variables rise or fall together. (−) Minus sign represents a negative correlation, where one variable increases as the other decreases.

**Table 6 jcm-13-07123-t006:** Correlations between IIEF, age, and prostate dimensions.

		IIEF-15	D1	D2	D3	Volume	Age
IIEF-15	Pearson correlation	1	−0.122 **	−0.054	−0.102 **	−0.118 **	−0.471 **
	Sig. (two-tailed-*p* value)		0.000	0.098	0.002	0.000	0.000
Volume	Pearson correlation	−0.12 **	0.744 **	0.792 **	0.836 **	1	0.317 **
	Sig. (two-tailed-*p* value)	0.000	0.000	0.000	0.000		0.000
Age	Pearson correlation	−0.47 **	0.256 **	0.221 **	0.287 **	0.317 **	1
	Sig. (two-tailed-*p* value)	0.000	0.000	0.000	0.000	0.000	

** Correlation is significant at the 0.01 level (two-tailed). (+) Plus sign in the correlation coefficient indicates a positive correlation, meaning both variables rise or fall together. (−) Minus sign represents a negative correlation, where one variable increases as the other decreases.

**Table 7 jcm-13-07123-t007:** The presence of cardiovascular disease is statistically significantly correlated with the moderate and severe LUTS symptoms.

*p*-0.0001			IPSS GROUPS		
		≤7	(8–19)	≥20	Total
**Cardiovascular diseases (yes)**	Number	309	294	123	726
	% raw	25.9%	49.7%	24.4%	100.0%
	% column	14.2%	25.0%	28.1%	21.3%
**Total**	Number	360	392	171	923
	% raw	39.0%	42.5%	18.5%	100.0%
	% column	100.0%	100.0%	100.0%	100.0%

**Table 8 jcm-13-07123-t008:** The presence of cardiovascular disease is statistically significantly correlated with IIEF < 20.

*p*-0.0001			IIEF GROUPS			
		IIEF = 61–75	IIEF = 41–60	IIEF = 21–40	IIEF < 20	Total
Cardiovascular diseases (yes)	Number	17	86	41	53	197
	% raw	8.6%	43.7%	20.8%	26.9%	100.0%
	% column	12.9%	20.4%	19.2%	34.0%	21.3%
Total	Number	132	421	214	156	923
	% raw	14.3%	45.6%	23.2%	16.9%	100.0%
	% column	100.0%	100.0%	100.0%	100.0%	100.0%

**Table 9 jcm-13-07123-t009:** The multivariate regression model: IPSS is positively influenced by age, D3, and prostate volume.

Dependent Y			IPSS Multiple Regression		
Independent Variables	Coefficient	Std. Error		t	*p*
Age	0.1082	0.03645		2.969	0.0031
D1	−0.03097	0.08005		−0.387	0.6990
D2	−0.03992	0.07971		−0.501	0.6167
D3	0.1333	0.04121		3.235	0.0013
PSA	0.1670	0.1195		1.397	0.1627
Volume	0.05380	0.01771		3.039	0.0024

**Table 10 jcm-13-07123-t010:** The multivariate regression model: IIEF is positively influenced by age, D3, and prostate volume.

Dependent Y			IIEF Multiple Regression		
Independent Variables	Coefficient	Std. Error		t	*p*
Age	−0.9717	0.06217		−15.629	<0.0001
D1	−0.04968	0.1364		−0.364	0.7158
D2	0.1133	0.1362		0.832	0.4059
D3	−0.2482	0.07964		−3.117	0.0019
PSA	0.1486	0.2056		0.723	0.4699
Volume	−0.1231	0.03422		−3.598	0.0003

## Data Availability

The research data that support the findings are not publicly available. The dataset is available on request from the authors in accordance with the hospital rules, patients’ consent, and local ethics committee.

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
