# Peer review of "Prostate Dimensions and Their Impact on LUTS and Erectile Function: Is Length the Missing Link?"

_jcm, 2024, doi:10.3390/jcm13237123_

Round 1

Reviewer 1 Report

Comments and Suggestions for Authors

Dear authors:

You conducted a study with an excellent research topic. It is a retrospective multicenter study using data from about 17 years.

1. The title only mentions prostate dimensions, but the main text and Discussion list the correlations with systemic diseases such as CVD and DM, which is distracting. I think it would be better to boldly delete these parts and focus on the originality of the study.

2. It is well known that severe IPP leads to high IPSS, severe voiding dysfunction, and low drug effects. What does it mean that only Length (D3) has a statistically significant correlation with IPSS in this study? Is there a positive relationship even when IPP is not severe? Or is it simply due to IPP? I think it would be good to analyze these two separately.

3. What is the explanation for the conclusion that there is a correlation between prostate volume and IIEF, and also a correlation between D3 and IIEF? In reference 50, you mentioned, there is only a phenomenon, but no hypothesis to explain it. It would be good if you could mention what mechanism the authors think it is.

4. The references for the IIEF changes after TURP, i.e., after the transition zone is removed, cited at the end of the Discussion to explain the correlation between prostate volume and IIEF are somewhat sparse and not convincing. If you think it is meaningful, it would be better to supplement it a bit more.

Many thanks for your hard works.

Author Response

Dear Reviewer,
Thank you for your kind words and for the time and expertise you dedicated to reviewing our article, "Prostate Dimensions and Their Impact on LUTS and Erectile Function: Is Length the Missing Link?" We greatly appreciate your positive feedback regarding our research topic and methodology, particularly noting the value of a retrospective multicenter study with data spanning approximately 17 years.
Your comments and suggestions are highly valuable to us. They not only provide insight but also help us refine the clarity and rigor of our work. We are grateful for your attention to detail, as your observations highlight areas for improvement that will strengthen our study’s impact and ensure that our findings contribute meaningfully to the existing literature.
Please find our responses to each of your comments and suggestions below. We have taken each one seriously and made adjustments in the manuscript where needed to address your recommendations thoughtfully.
Thank you again for your invaluable feedback and for your role in the refinement of our study.
Warm regards,
The Authors

Comments:1.The title only mentions prostate dimensions, but the main text and Discussion list the correlations with systemic diseases such as CVD and DM, which is distracting. I think it would be better to boldly delete these parts and focus on the originality of the study.

Answer:I appreciate your insight regarding the potential streamlining of the manuscript by omitting sections on correlations with systemic diseases such as CVD and DM.

However, I find it challenging to delete these parts for several reasons. First, for patients with LUTS and ED, the roles of CVD and DM are still widely discussed in the literature and omitting them from the discussion might raise questions about why we chose not to address these conditions, especially as they were part of our study parameters. This also applies to factors like smoking and alcohol consumption, which we evaluated as part of the broader context of these urological and systemic health issues.

Secondly, and honestly, I hesitate to “boldly delete” these sections as suggested, as I feel this would alter the structure and completeness of the manuscript as initially conceived. These aspects were carefully considered to provide a more comprehensive understanding of the multi-faceted relationships involved, as we sought to capture the full scope of contributing factors and their potential implications.

Comments: 2. It is well known that severe IPP leads to high IPSS, severe voiding dysfunction, and low drug effects. What does it mean that only Length (D3) has a statistically significant correlation with IPSS in this study? Is there a positive relationship even when IPP is not severe? Or is it simply due to IPP? I think it would be good to analyze these two separately.

Answer: Thank you for raising this insightful question. Indeed, the relationship between intravesical prostatic protrusion (IPP) and IPSS is well-established. The degree of IPP can be measured from the intravesical edge of the prostate to the base of the bladder in the mid-sagittal abdominal ultrasound. However, IPP measurement was not initially considered as part of this study's design, though this assessment was later performed for some patients. As a result, we did not evaluate the impact of IPP on IPSS statistically and, consequently, cannot determine whether there is a positive relationship even when IPP is not severe. We recognize that this omission limits our findings, and we consider it one of the study's limitations. Thank you for highlighting this important aspect, which may be a valuable consideration for future research on this topic. I have included these remarks in the manuscript, acknowledging this limitation of the study. Linie 353-364.

Comments: 3. What is the explanation for the conclusion that there is a correlation between prostate volume and IIEF, and also a correlation between D3 and IIEF? In reference 50, you mentioned, there is only a phenomenon, but no hypothesis to explain it. It would be good if you could mention what mechanism the authors think it is.

Answer: 

  1. In the presented data, there is a significant relationship between D1, D3, and IIEF. Considering that prostate volume was calculated using the formula Prostate Volume (ml) ≈ 0.52 × D1 × D2 × D3, there is a strong likelihood that volume would statistically correlate with IIEF, even though D2 does not show a significant correlation with IIEF.
  2. I have added several paragraphs to the article discussing the hypotheses we considered regarding the possible physiological and pathophysiological mechanisms by which D3 correlates with IIEF and IPSS. Line :379-396

Comments: 4. The references for the IIEF changes after TURP, i.e., after the transition zone is removed, cited at the end of the Discussion to explain the correlation between prostate volume and IIEF are somewhat sparse and not convincing. If you think it is meaningful, it would be better to supplement it a bit more.

Answer: Thank you for your comment. Indeed, in the case of TURP, additional mechanisms may be involved in the etiology of potential ED. Since the role of TURP was not evaluated in this article, including it as an argument is, as you pointed out, sparse and unconvincing. Therefore, that paragraph, along with its bibliographic references, has been removed from the manuscript.

Reviewer 2 Report

Comments and Suggestions for Authors

Dear Authors,

I read with interest your article, however I found some points that needs to be further elucidated:

- IPSS and IIEF are natural number, with no decimals, thus they should all be changed according to it

- In your article you found an association between D3 and IIEF and IPSS, but only at pearson and spearman test, a multivariate analysis is deeply welcome, alike a multinomial logistic regression to complete the study

- I found that not all the patients have all the variables. Patients with incomplete data set should be excluded from the analysis

- The discussion completely lacks a strength and limitations section of your study. It must be added

Author Response

Dear Reviewer,

Thank you sincerely for the time, effort, and professionalism you invested in evaluating our article, "Prostate Dimensions and Their Impact on LUTS and Erectile Function: Is Length the Missing Link?" Your comments and suggestions were both insightful and constructive, highlighting essential areas for improvement.

We agree that the statistical analysis required refinement, particularly regarding patients with incomplete data and the inclusion of additional statistical analyses. We have now made these adjustments in line with your recommendations to ensure greater accuracy and comprehensiveness in our findings. Additionally, we have introduced a Strengths and Limitations section to address the study's scope and boundaries explicitly.

I will respond in detail to each of your comments below, outlining the specific changes made in response to your feedback.

Thank you once again for your valuable input, which has helped significantly to enhance the quality of our work.

Warm regards,
The Authors

Comments: IPSS and IIEF are natural number, with no decimals, thus they should all be changed according to it

Answer: I have updated the data where IPSS and IIEF values were displayed with decimals: Table 1, Table 4.

Comments: In your article you found an association between D3 and IIEF and IPSS, but only at pearson and spearman test, a multivariate analysis is deeply welcome, alike a multinomial logistic regression to complete the study.

Answer: Multivariate analysis was performed using linear regression. The dependent variables were IPSS or IIEF, while the independent variables included age, D1, D2, D3, PSA, and prostate volume. These analyses have been incorporated into the manuscript: line 153-155, Table 9, Table 10.

Comments: I found that not all the patients have all the variables. Patients with incomplete data set should be excluded from the analysis

Answer: I have revised the exclusion criteria as per the requirements – Lines 137-138.

Additionally, I have updated the statistical data throughout the entire article to reflect the final count of 923 patients who met all study requirements. These modifications are reflected in Table 1, Table 2, Table 3, Table 4, as well as in Lines 187-189.

Comments: The discussion completely lacks a strength and limitations section of your study. It must be added.

Answer: 

I have included the following section to highlight the strengths and limitations of our study. Line 397-406.
“The strengths of this study include a large patient sample, an innovative approach to correlating specific prostate dimensions with both IIEF and IPSS scores, and the exami-nation of IPSS and IIEF changes across age groups. Additionally, it evaluates other key parameters, such as cardiovascular disease and diabetes, as well as lifestyle risk factors like smoking and alcohol consumption.
The main limitations involve the use of abdominal ultrasound for prostate measurement, which is less precise than transrectal ultrasound; however, abdominal ultrasound was chosen for consistency throughout the study. 
Another limitation is the lack of detailed erectile function analysis based on the first five questions of the IIEF, as only the IIEF-15 total score was included in the dataset, limiting a more specific correlation.”

Round 2

Reviewer 2 Report

Comments and Suggestions for Authors

Dear Authors,

thank you for modifying the article according to my observations.

I found really interesting the results of multivariate analysis, especially of the prostate craniocaudal extension, which also in my clinical experience is deeply related to IPSS. That because the median lobe is comprised in that diameter, thus affecting LUTS burden

I have no other requests, but I wanted to highlight that in Table 4 PSA has been  removed, but it is still reported in Table title. You should decide wheter to keep it or not

Author Response

Comments:

Dear Authors,

thank you for modifying the article according to my observations.

I found really interesting the results of multivariate analysis, especially of the prostate craniocaudal extension, which also in my clinical experience is deeply related to IPSS. That because the median lobe is comprised in that diameter, thus affecting LUTS burden

I have no other requests, but I wanted to highlight that in Table 4 PSA has been  removed, but it is still reported in Table title. You should decide wheter to keep it or not

Response:

Dear Reviewer,

Thank you for your valuable feedback and for highlighting the clinical relevance of the multivariate analysis, particularly regarding the prostate craniocaudal extension and its association with IPSS. We greatly appreciate your insight and are pleased to know that our findings align with your clinical experience.

Regarding Table 4, we have addressed your observation by removing PSA from the table title to ensure consistency with the data presented. Thank you for bringing this to our attention.

We sincerely appreciate your thorough review and constructive comments, which have greatly contributed to improving the quality of our manuscript.

Best regards,

Authors